# In Vivo Multimodal Imaging of Stem Cells Using Nanohybrid Particles Incorporating Quantum Dots and Magnetic Nanoparticles

**DOI:** 10.3390/s22155705

**Published:** 2022-07-30

**Authors:** Shota Yamada, Hiroshi Yukawa, Kaori Yamada, Yuki Murata, Jun-ichiro Jo, Masaya Yamamoto, Ayae Sugawara-Narutaki, Yasuhiko Tabata, Yoshinobu Baba

**Affiliations:** 1Department of Energy Engineering, Graduate School of Engineering, Nagoya University, Furo-cho, Chikusa-ku, Nagoya 464-8603, Japan; yamada.shota.b6@s.mail.nagoya-u.ac.jp (S.Y.); ayae@energy.nagoya-u.ac.jp (A.S.-N.); 2Institute of Nano-Life-Systems, Institutes of Innovation for Future Society, Nagoya University, Furo-cho, Chikusa-ku, Nagoya 464-8603, Japan; kaori040624@gmail.com (K.Y.); babaymtt@chembio.nagoya-u.ac.jp (Y.B.); 3Department of Biomolecular Engineering, Graduate School of Engineering, Nagoya University, Furo-cho, Chikusa-ku, Nagoya 464-8603, Japan; 4Institute for Quantum Life Science, Quantum Life and Medical Science Directorate, National Institutes for Quantum Science and Technology, Anagawa 4-9-1, Inage-ku, Chiba 263-8555, Japan; 5B-3Frontier, Advanced Analytical and Diagnostic Imaging Center (AADIC)/Medical Engineering Unit (MEU), Institute for Advanced Research, Nagoya University, Tsurumai 65, Showa-ku, Nagoya 466-8550, Japan; 6Department of Medical-Engineering Collaboration Supported by SEI Group CSR Foundation, Nagoya University, Tsurumai 65, Showa-ku, Nagoya 466-8550, Japan; 7Department of Regeneration Science and Engineering, Institute for Frontier Life and Medical Sciences, Kyoto University, 53 Kawara-cho Shogoin, Sakyo-ku, Kyoto 606-8507, Japan; y.murata406@gmail.com (Y.M.); jo@frontier.kyoto-u.ac.jp (J.-i.J.); yasuhiko@infront.kyoto-u.ac.jp (Y.T.); 8Department of Metallurgy, Materials Science and Materials Processing, Graduate School of Engineering, Tohoku University, Aoba-yama 02, Aoba-ku, Sendai 980-8579, Japan; masaya@material.tohoku.ac.jp

**Keywords:** magnetic nanoparticles, quantum dots (QDs), in vivo imaging, adipose tissue-derived stem cells (ASCs)

## Abstract

The diagnosis of the dynamics, accumulation, and engraftment of transplanted stem cells in vivo is essential for ensuring the safety and the maximum therapeutic effect of regenerative medicine. However, in vivo imaging technologies for detecting transplanted stem cells are not sufficient at present. We developed nanohybrid particles composed of dendron-baring lipids having two unsaturated bonds (DLU2) molecules, quantum dots (QDs), and magnetic nanoparticles in order to diagnose the dynamics, accumulation, and engraftment of transplanted stem cells, and then addressed the labeling and in vivo fluorescence and magnetic resonance (MR) imaging of stem cells using the nanohybrid particles (DLU2-NPs). Five kinds of DLU2-NPs (DLU2-NPs-1-5) composed of different concentrations of DLU2 molecules, QDs525, QDs605, QDs705, and ATDM were prepared. Adipose tissue-derived stem cells (ASCs) were labeled with DLU2-NPs for 4 h incubation, no cytotoxicity or marked effect on the proliferation ability was observed in ASCs labeled with DLU2-NPs (640- or 320-fold diluted). ASCs labeled with DLU2-NPs (640-fold diluted) were transplanted subcutaneously onto the backs of mice, and the labeled ASCs could be imaged with good contrast using in vivo fluorescence and an MR imaging system. DLU2-NPs may be useful for in vivo multimodal imaging of transplanted stem cells.

## 1. Introduction

Stem cell transplantation therapy were known as very simple and low invasive medicine in regenerative medicine and could be utilized for treating many serious diseases such as heart, liver, and central nervous system (CNS) disorders. Indeed, stem cell transplantation therapy with somatic stem cells which can transplant with lower invasiveness, such as bone marrow stem cells (BMSCs) [1,2] and adipose tissue-derived stem cells (ASCs) [3,4], has been applied in clinical practice. It is known that the accumulation and engraftment of transplanted stem cells in affected tissues and organs strongly influences the therapeutic efficacy [5,6], and the inflammatory state of the affected tissues and organs is also considered to be similarly affected [7,8,9], but little is known on this subject.

Therefore, in vivo real-time imaging of the kinetics of transplanted stem cell behavior, accumulation, and engraftment is essential to ensure maximum therapeutic safety and efficacy of stem cell transplantation. Several methods have been used in clinical practice, such as X-ray computed tomography (CT) and magnetic resonance imaging (MRI); however, these methods struggle to detect transplanted stem cells with high sensitivity when these modalities are used alone. To overcome these issues, multimodal imaging combining fluorescence imaging (FI) and MRI that can detect the small numbers of transplanted stem cells has been developed and has drawn a great deal of attention [10,11,12].

Quantum dots (QDs) have several outstanding fluorescence properties to conventional organic labels such as a high luminance, resistance to photobleaching (long time labeling), wide range of excitation wavelength, and narrow fluorescence wavelength. In particular, QDs which are water-soluble and emit fluorescence in the near-infrared (NIR) region, have attracted attention for their diagnostic applications in the medical field as useful fluorescent probes [13,14,15,16,17,18,19,20]. We previously developed stem cell labeling technology using QDs and in vivo fluorescence imaging technology of transplanted stem cells labeled with QD [4,21,22,23,24].

Magnetic nanoparticles are known as contrast agents for MRI. Gadolinium (Gd) and superparamagnetic iron oxide (SPIO) nanoparticles are generally used as magnetic nanoparticles in order to increase the contrast of issues in typical imaging studies [25,26,27,28]. Various SPIO nanoparticles have been developed as contrast agents; ferucarbotran (Resovist), an anionic SPIO nanoparticle with a carboxydextran coating, has been successfully applied in the clinical setting as a liver contrast agent [29,30,31,32]. We observed the alkali-treated dextrancoated magnetic iron oxide nanoparticles (ATDM), which are a major component of ferucarbotran (Resovist), and have already applied magnetic nanoparticles to cell labeling and in vivo MRI techniques [33,34].

In this study, we developed nanohybrid particles (DLU2-NPs) composed of DLU2 molecules, three kinds of QDs, and magnetic nanoparticles (ATDM) and addressed the labeling and in vivo fluorescence and MR imaging of transplanted stem cells labeled with the DLU2-NPs.

## 2. Materials and Methods

### 2.1. Materials

QDs525, QDs605, QDs705, Hank’s balanced salt solution, phosphate-buffered saline (PBS), Dulbecco’s modified Eagle’s medium and full-length name of F12 (DMEM/F12), penicillin/streptomycin, and HEPES buffer were purchased from Life Technologies^TM^ Japan (Tokyo, Japan). Type I collagenase was purchased from Koken Co., Ltd. (Tokyo, Japan). ATDM was purchased from Cosmo Bio*^®^* Japan (Tokyo, Japan). Fetal bovine serum (FBS) was purchased from Trace Scientific Ltd. (Melbourne, Australia). Cell Counting Kit-8 (CCK-8) was purchased from DOJINDO Laboratories (Kumamoto, Japan).

### 2.2. Animals

C57BL/6 mice were purchased from Japan SLC (Hamamatsu, Japan). The mice were kept in a controlled environment (12 h light/dark cycles at 21 °C) with free access to water and a standard chow diet before sacrifice. All conditions and handing of animals in this study were in accordance with the protocols approved by the Nagoya University Committee on Animal Use and Care.

### 2.3. Isolation and Culture of ASCs

The isolation and culture of ASCs have been reported previously [4]. Female C57BL/6 mice 7–14 months of age were killed by cervical dislocation; adipose tissue specimens in the inguinal groove were isolated and washed extensively with Hank’s balanced salt solution or PBS to remove the blood cells. The isolated adipose tissue specimens were cut finely and digested with 1 mL of 1 mg/mL type I collagenase (274 U/mg) at 37 °C in a shaking water bath for 45 min. The cells were filtrated using 250 µm nylon cell strainers and suspended in DMEM/F12 containing 20% FBS, and 100 U/mL of penicillin/streptomycin (culture medium). They were centrifuged at 1200 rpm for 5 min at room temperature and ASCs were obtained from the pellet. The cells were then washed three times by suspension and centrifugation in culture medium and then were incubated overnight in culture medium at 37 °C with 5% CO_2_. The primary cells were cultured for several days until they reached confluence and defined as passage “0”. The cells were used for the experiments between passages 2 and 5.

### 2.4. Preparation of DLU2-NPs

DLU2 was dissolved in chloroform/methanol (1/1, *v*/*v*) in a volumetric flask and the concentration of DLU2 was adjusted to 10 mg/mL. DLU2 solution was distilled away under nitrogen purging and dried by vacuum drying. The solutions of QDs525 (100 nM), QDs605 (100 nM), QDs705 (100 nM), and ATDM (500 µg/mL) were added to the lipid film composed of DLU2, and then 10 mM HEPES buffer was added. After sonication of the mixture for 30 min, DLU2-NPs (10 mg/mL of DLU2) were obtained.

### 2.5. Measurement of Particle Size and Zeta Potential of DLU2-NPs

DLU2-NPs were diluted 1:100 using PBS, and the particle size and zeta potential were measured by dynamic light scattering (DLS) using the Zetasizer Nano ZS (Malvern Instruments, Ltd., Herrenberg, Germany). These measurements were performed at 25 °C.

### 2.6. Confocal Microscopy Observation of Labeled ASCs

ASCs (1 × 10^5^ cells) were seeded in 35 mm glass-bottom dishes with 200 µL of culture medium for 24 h and that was replaced with 200 µL of DLU2-NPs diluted 640-fold in medium. After incubation for 1 or 4 h, ASCs were washed twice with medium. The labeled ASCs were observed with a high-speed multi-photon confocal laser microscope (A1R MP^+^; Nikon Corporation, Tokyo, Japan).

### 2.7. Cytotoxicity of DLU2-NPs to ASCs

ASCs (1 × 10^4^ cells) were seeded in 96-well plates (BD Biosciences) with 100 µL of culture medium that was then replaced with 100 µL of DLU2-NPs diluted 640-, 320-, 160-, 80-, or 40-fold in medium. After incubation for 4 h, ASCs were washed twice with medium. Viable cells were counted by using a CCK-8. CCK-8 reagent (10 µL) was added to each well, and the reaction was allowed to proceed for 1 h. The absorbance of the sample at 450 nm was measured against a background control using a microplate reader (PLARstar OPTIMA, BGM Labtech, Ortenberg, Germany).

### 2.8. Proliferation test of ASCs Labeled with DLU2-NPs

ASCs (2 × 10^3^ cells) were seeded in 96-well plates and labeled with 100 µL of DLU2-NPs diluted 640- or 320-fold in medium in the same way. After incubation for 4 h, ASCs were washed twice with medium. Viable cells were counted using a CCK-8 in the same way.

### 2.9. In Vitro Fluorescence and MR Imaging of ASCs Labeled with DLU2-NPs

ASCs were cultured for several days until they reached confluence and then were replaced with 5 mL of DLU2-NPs diluted 640-, 320-, or 160-fold with medium. After incubation for 4 h, ASCs were washed twice with medium and collected in 1.5 mL centrifuge tubes. The fluorescence images of ASCs with DLU2-NPs in 1.5 mL centrifuge tubes were taken using the IVIS Lumina K Series III (PerkinElmer Inc, Waltham, MA, USA; excitation filter: 460–620 nm, emission filter: 520, 570, and 710 nm long-pass). In vitro MR images of ASCs with DLU2-NPs in 1.5 mL centrifuge tubes were taken using the MR VivoLVA (DS Pharma Biomedical Co., Ltd., Osaka, Japan). Regarding the spin echo, the images were obtained with repetition time (TR) = 500.0 ms and echo time (TE) = 9.0 ms, or TR = 2000.0 ms and TE = 69.0 ms and field of view = 60 × 60 mm.

### 2.10. In Vivo Fluorescence and MR Imaging of ASCs Labeled with DLU2-NPs

ASCs were cultured for several days until they reached confluence, and then were replaced with 5 mL of DLU2-NPs diluted 640- or 320-fold in medium. After incubation for 4 h, ASCs were washed twice. ASCs (3 × 10^6^ cells) labeled with DLU2-NPs with 0.2 mL PBS were transplanted into the space of back subcutaneously of C57BL/6 mice. In vivo fluorescence imaging of ASCs labeled with DLU2-NPs images were taken using the IVIS Lumina K Series III (PerkinElmer Inc.; excitation filter: 500–620 nm, emission filter: 520, 620 and 710 nm long-pass). In vivo MRI were taken using the MR VivoLVA (DS Pharma Biomedical Co.). Regarding the spin echo, the images were obtained with repetition time (TR) = 500.0 ms and echo time (TE) = 9.0 ms, or TR = 2000.0 ms and TE = 69.0 ms and field of view = 60 × 60 mm.

### 2.11. Statistical Analyses

Numerical values are presented as the mean ± standard deviation (SD). Each experiment was repeated three times. Statistical significance was evaluated using an unpaired Student’s *t*-test for comparisons between the two groups; *p*-values < 0.05 were considered statistically significant. All statistical analyses were performed using the SPSS software package.

## 3. Results and Discussion

### 3.1. Properties of DLU2-NPs

The schematic diagram of DLU2-NPs composed of three kinds of QDs (QDs525, 605, 705), magnetic nanoparticles (ATDM), and DLU2 molecules is shown in Figure 1a. The chemical structural formula of the DLU2 molecule consisted of a cationic lipid composed of a polyamidoamine dendron and two alkyl chains was shown in Figure 1b.

The two alkyl chains in the DLU2 molecule are said to have a membrane fusion function, and the presence of an unsaturated bond further promotes membrane fusion. These effects are expected to promote the introduction of quantum dots and magnetic nanoparticles in liposomes into cells [23]. The tertiary amino groups of the polyamidoamine dendron have been reported to adsorb protons, thereby reducing the pH endosomes and accelerating the influx of H^+^ and Cl^-^, causing Cl^-^ accumulation and permeation swelling/dissolution [35].

The concentrations of DLU2 molecules, QDs, and ATDM contained in DLU2-NPs are shown in Table 1. The distribution of the particle diameter and zeta potential of DLU2-NPs is shown in Figure 1c,d. The average particle diameter and zeta potential of DLU2-NPs were 135.7 nm and 28.7 mV, respectively.

### 3.2. ASCs labeling by DLU2-NPs

DLU2-NPs were transduced into ASCs for 4 h of incubation at 37 °C (Figure 2a). The morphology and fluorescence images of ASCs were obtained by confocal microscopy. The strong fluorescence derived from QDs525, QDs605, and QDs705 was detected at the same site in the cytoplasm of ASCs (Figure 2b–g). No abnormalities in the morphology of labeled ASCs were observed. These data suggest that ASCs can be labeled with DLU2-NPs by simple culture for 4 h without cytotoxicity.

### 3.3. Cytotoxicity of DLU2-NPs to ASCs and the Proliferation Rate of ASCs Labeled with DLU2-NPs

To examine the cytotoxicity of DLU2-NPs to ASCs, ASCs were transduced with various concentrations (640-, 320-, 160-, 80-, or 40-fold diluted solution) of DLU2-NPs for 4 h, and ASCs were incubated for 24 h. Significant cytotoxicity was observed in the ASCs labeled with 160-, 80-, and 40-fold-diluted solution of DLU2-NPs; however, no cytotoxicity was observed in the ASCs labeled with 640- and 320-fold- diluted solution of DLU2-NPs (Figure 3a).

The influence of the DLU2-NPs on the proliferation ability of ASCs was also examined at non-cytotoxic concentrations. The proliferation rates of ASCs labeled with DLU2-NPs in non-cytotoxic concentrations were confirmed to be almost equal to that of normal ASCs (Figure 3b). These data suggest that DLU2-NPs have no cytotoxicity and no effect on the proliferation ability of ASCs at 640- and 320-fold-diluted concentrations.

### 3.4. In Vitro Fluorescence and MR Imaging of ASCs Labeled with DLU2-NPs

To examine the detectable labeling concentration of DLU2-NPs, ASCs labeled with various concentrations of DLU2-NPs were collected in PBS and spun down. The pellets of ASCs in microtubes were then prepared for a fluorescence analysis (Figure 4a). All fluorescence derived from three kinds of QDs (525, 605 and 705) were detected even at 640-fold-diluted concentrations (Figure 4b–d). The ASC pellets were also then prepared for MR imaging in microtubes (Figure 4e). The MR signal of ASCs labeled with DLU2-NPs were detected on T2-weighted imaging even at 640-fold-diluted concentrations (Figure 4f,g). In addition, the MR signal (drawn by yellow dot line) of ASCs labeled with DLU2-NPs was evaluated (Figure 4h). These data suggest that ASCs labeled with DLU2-NPs could be detected with multicolor fluorescence and MR imaging.

### 3.5. In Vivo Fluorescence and MR Imaging of Transplanted ASCs Labeled with DLU2-NPs

ASCs (3 × 10^6^ cells) labeled with DLU2-NPs were subcutaneously transplanted with PBS onto the back of mice (Figure 5a). All three kinds of QDs (QDs525, QDs605, and QDs705) were observed by in vivo fluorescence imaging system (IVIS Lumina K Series III) (Figure 5b–d). The fluorescence derived from QDs705 was detected with high efficiency, as the fluorescence in the near infrared (NIR) (700–900 nm) known as the biological window. In contrast, the MR signal of ATDM was observed using an in vivo MR imaging system (MR VivoLVA) shown in a yellow-dotted circle (Figure 5e–l). These data suggest that transplanted ASCs labeled with DLU2-NPs in mice could be detected with both fluorescence and MR imaging.

## 4. Conclusions

In summary, multifunctional nanoparticles (DLU2-NPs) were prepared and examined for their utility in stem cell labeling and in vivo imaging of transplanted stem cells. DLU2-NPs were able to be transduced into ASCs with high efficiency by simple incubation for 4 h. No cytotoxicity of ASCs labeled with DLU2-NPs was noted under certain concentrations of DLU2-NPs. In addition, no effect of DLU2-NPs on the proliferation ability of labeled ASCs was observed. The fluorescence and MR signal of ASCs labeled with DLU2-NPs was detected in vitro. Furthermore, in vivo fluorescence and MR multimodal imaging of transplanted ASCs labeled with DLU2-NPs was achieved using in vivo imaging systems. These data suggest that the development of new multimodal imaging technologies able to detect the dynamics of transplanted stem cells may be imminent.

## Figures and Tables

**Figure 1 sensors-22-05705-f001:**
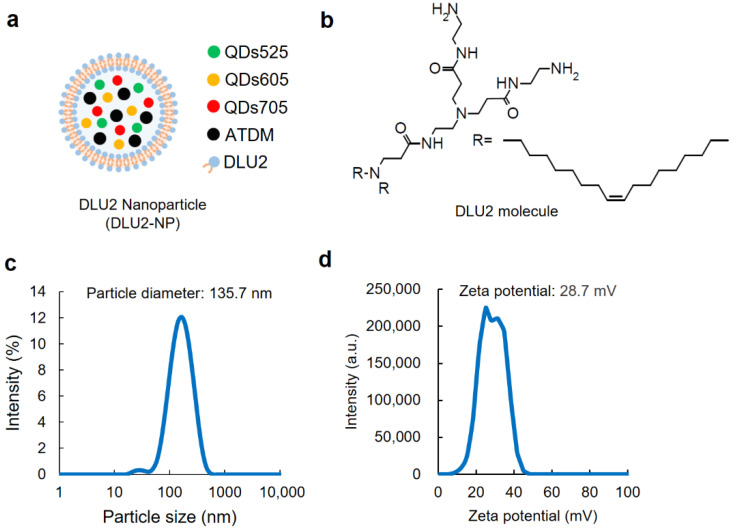
The properties of DLU2-NPs. A schematic diagram of DLU2-NPs (**a**) and DLU2 molecule (**b**). The particle diameter (**c**) and the zeta potential (**d**) of DLU2-NPs in PBS.

**Figure 2 sensors-22-05705-f002:**
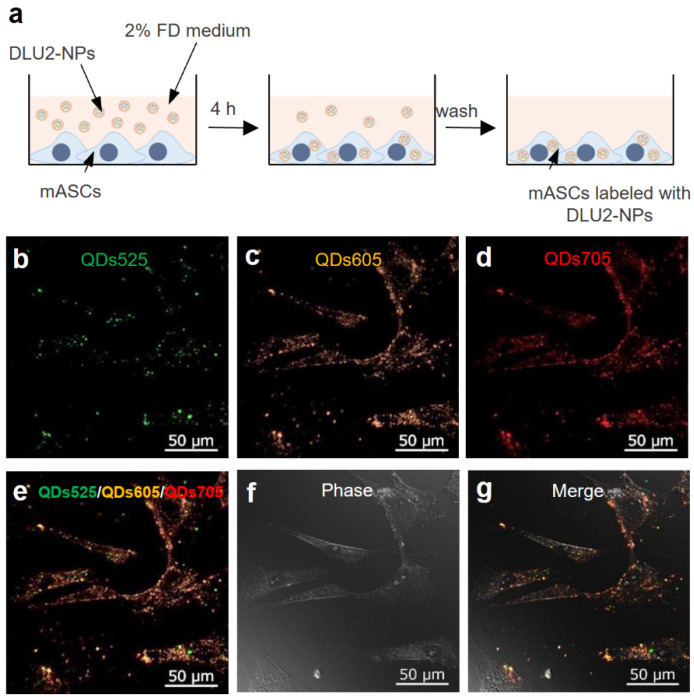
ASCs labeling by DLU2-NPs. A schematic diagram of ASCs labeling by DLU2-NPs (**a**). The confocal microscope images of ASCs labeled with DLU2-NPs (**b**–**g**).

**Figure 3 sensors-22-05705-f003:**
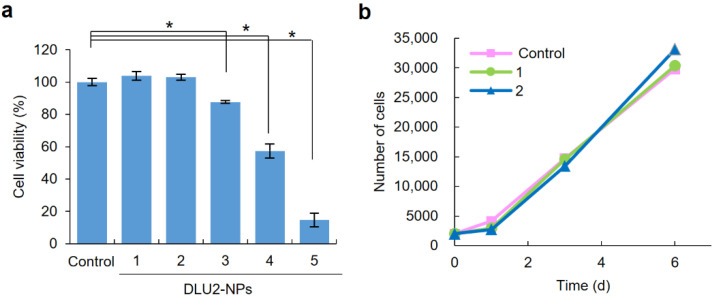
Cytotoxicity of DLU2-NPs to ASCs and the proliferation rate of ASCs labeled with DLU2-NPs. The cytotoxicity of DLU2-NPs (640-, 320-, 160-, 80-, 40-fold-diluted solution of DLU2-NPs) to ASCs (**a**), and the proliferation rate of ASCs labeled with DLU2-NPs (640-, 320-fold dilute solution of DLU2-NPs) (**b**). * *p* < 0.05.

**Figure 4 sensors-22-05705-f004:**
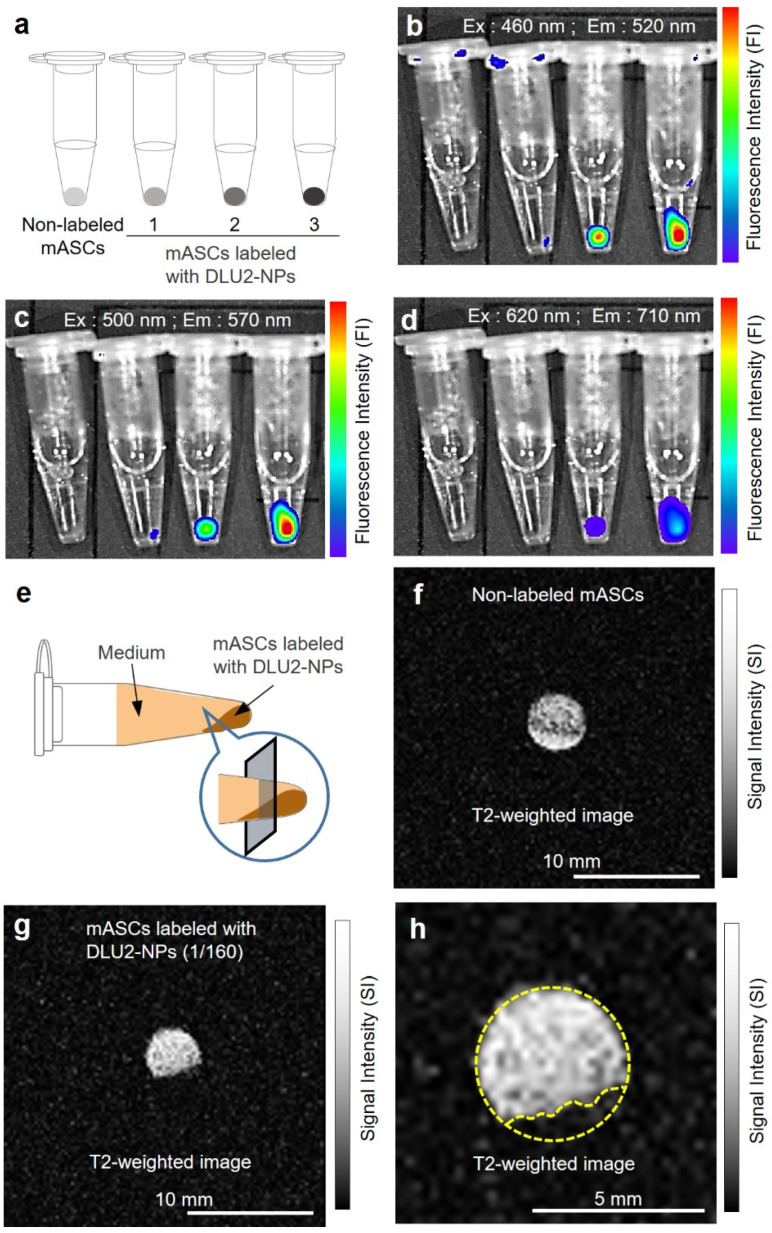
In vitro fluorescence and MR imaging of ASCs labeled with DLU2-NPs. In vitro fluorescence images of ASCs (3 × 10^6^ cells) labeled with various concentrations of DLU2-NPs (640, 320, 160-dilute solution), excitation: 460–620 nm, emission: 520–710 nm (**a**–**d**). In vitro MR imaging of ASCs labeled with DLU2-NPs (**e**–**h**).

**Figure 5 sensors-22-05705-f005:**
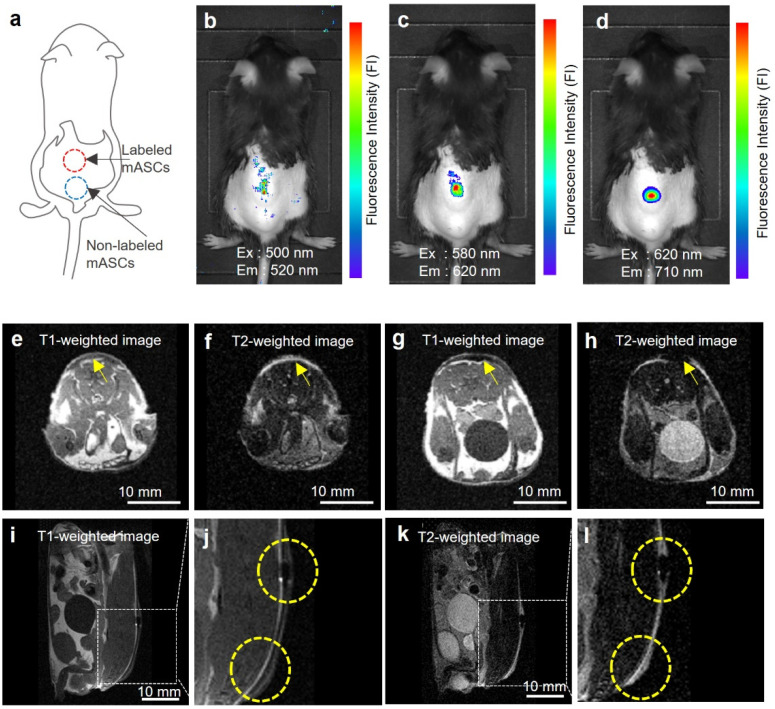
In vivo fluorescence and MR imaging of ASCs labeled with DLU2-NPs. In vivo fluorescence images of ASCs (3 × 10^6^ cells) labeled with DLU2-NPs (320-dilute solution) after subcutaneous transplantation onto the back of a mouse, excitation: 500–620 nm, emission: 520–710 nm (**a**–**d**). In vivo MR images of labeled ASCs, the transversal images of non-labeled ASCs (**e**,**f**), labeled ASCs (**g**,**h**), the sagittal images (**i**,**j**,**k,l**), T1-weighted image (**e**,**g**,**i,j**), T2-weighted images (**f**,**h**,**k,l**). (**j**,**l**) are enlarged images of (**i**,**k**). Yellow arrows and dotted circles indicate the locations where ASCs were transplanted.

**Table 1 sensors-22-05705-t001:** Information and dilution conditions of DLU2-NPs.

	Dilution factor	Lipid (mg/mL)	QDs525 (nM)	QDs605 (nM)	QDs705 (nM)	ATDM (μg/mL)
Stock Solution		10	100	100	100	500
1	640	0.016	0.16	0.16	0.16	0.78
2	320	0.031	0.31	0.31	0.31	1.6
3	160	0.063	0.63	0.63	0.63	3.1
4	80	0.13	1.3	1.3	1.3	6.3
5	40	0.25	2.5	2.5	2.5	13

## Data Availability

The data presented in this study are available upon request to the corresponding author.

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
