# Peer review of "In Vivo Multimodal Imaging of Stem Cells Using Nanohybrid Particles Incorporating Quantum Dots and Magnetic Nanoparticles"

_sensors, 2022, doi:10.3390/s22155705_

Round 1

Reviewer 1 Report

This paper reports a hybrid nanoparticle for stem cell labeling by fluorescence imaging and MRI in the living animals. Overall, the data can support the statements of the authors. Here are some questions to be addressed before this paper can be accepted for publication:

1) The abbreviation “DLU2” is used from the beginning. The full name should be provided when it first appeared in the manuscript.

2) The NP is used for fluorescence imaging and MRI. Is there any organic integration between these two modalities in the animal imaging?

3) 4 h incubation is used for cell labeling. The data about the optimization of incubation time should be provided.

4) In the Introduction part. It is stated that “in vivo real-time imaging and diagnosis of the kinetics of transplanted 53 stem cell behavior”. But in fact, the reported NP only can be used for labeling. The NP does not have any diagnosis ability.

5) In Abstract. There are some grammar errors. For example, “… in order to diagnosis of the dynamics … ” and “… the labeled ASCs were able to be imaged …”.

6) In Fig 1a, three different kinds of QDs were encapsulated in the NP, but no data to prove that all these QDs were indeed successfully loaded.

7) “Materials” part is missing in the experimental part.

8) QD-525 is not suitable for animal imaging at all, so why encapsulating this kind of QDs in the NP for animal imaging?

9) We can still see the PI’s comments left in the manuscript. For example, “This section may be divided by subheadings. It should provide a concise and precise description of the experimental results, their interpretation, as well as the experimental conclusions that can be drawn.”

Reviewer 2 Report

The idea of creating multifunctional therapeutic and diagnostic agents is of undoubted interest. This allows to expand the possibilities and reduce the pharmacological capacity on the patient. The manuscript will be of interest to a wide range of readers, including young scientists who are starting their career in science.

Correction

Page 6, Line 210-214, Fig.3A

To examine the cytotoxicity of DLU2-NPs to ASCs, ASCs were transduced with var- 210

ious concentrations (40-, 80-, 160-, 320-, and 640-fold diluted solution) of DLU2-NPs for 4 211

h, and ASCs were incubated for 24 h. Significant cytotoxicity was observed in the ASCs 212

labeled with more than 160-fold-diluted solution of DLU2-NPs, however, no cytotoxicity 213

was observed in the ASCs labeled with 320- and 640-fold- diluted solution of DLU2-NPs 214

(Figure 3a).

The interpretation of Figure 3A is completely unacceptable. Cell viability decreases as seen in the figure at dilution 1-5 (40-, 80-, 160-, 320-, and 640-fold diluted solution) to values of 20%. This is a misrepresentation. Authors should reformat the figure and give a more detailed description of it. An explanation of the abbreviation SD is also missing.

The positive charge of the resulting nanoparticles can cause nonspecific interaction with negatively charged cells, as occurs with cationic transfectants. This leads to their high cytotoxicity at high concentrations.

After proofreading, the manuscript can be accepted for publication.
